# Severe T cell hyporeactivity in ventilated COVID-19 patients correlates with prolonged virus persistence and poor outcomes

Kerstin Renner[1], Tobias Schwittay[1], Sophia Chaabane[1], Johanna Gottschling[1], Christine Müller[1], Charlotte Tiefenböck[1], Jan-Niklas Salewski[1], Frederike Winter[1,2], Simone Buchtler[1], Saidou Balam[1], Maximilian V. Malfertheiner[3], Matthias Lubnow[3], Dirk Lunz[4], Bernhard Graf[4], Florian Hitzenbichler[5], Frank Hanses[5], Hendrik Poeck[6], Marina Kreutz[6], Evelyn Orsó[7], Ralph Burkhardt[7], Tanja Niedermair[8,9], Christoph Brochhausen[8,9], André Gessner[10], Bernd Salzberger[5] & Matthias Mack[1,2✉]

Coronavirus disease 2019 (COVID-19) can lead to pneumonia and hyperinflammation. Here we show a sensitive method to measure polyclonal T cell activation by downstream effects on responder cells like basophils, plasmacytoid dendritic cells, monocytes and neutrophils in whole blood. We report a clear T cell hyporeactivity in hospitalized COVID-19 patients that is pronounced in ventilated patients, associated with prolonged virus persistence and reversible with clinical recovery. COVID-19-induced T cell hyporeactivity is T cell extrinsic and caused by plasma components, independent of occasional immunosuppressive medication of the patients. Monocytes respond stronger in males than females and IL-2 partially restores T cell activation. Downstream markers of T cell hyporeactivity are also visible in fresh blood samples of ventilated patients. Based on our data we developed a score to predict fatal outcomes and identify patients that may benefit from strategies to overcome T cell hyporeactivity.

[1] Department of Nephrology, University Hospital Regensburg, Regensburg, Germany. [2] Regensburg Center for Interventional Immunology, Regensburg, Germany. [3] Department of Internal Medicine II, University Hospital Regensburg, Regensburg, Germany. [4] Department of Anesthesiology, University Hospital Regensburg, Regensburg, Germany. [5] Department of Infection Prevention and Infectious Diseases, University Hospital Regensburg, Regensburg, Germany. [6] Department of Internal Medicine III, University Hospital Regensburg, Regensburg, Germany. [7] Institute of Clinical Chemistry and Laboratory Medicine, University Hospital Regensburg, Regensburg, Germany. [8] Institute of Pathology, University of Regensburg, Regensburg, Germany. [9] Central Biobank Regensburg, University and University Hospital, Regensburg, Germany. [10] Institute Clinical Microbiology and Hygiene, University Hospital Regensburg, Regensburg, Germany. ✉email: matthias.mack@klinik.uni-regensburg.de

SARS-CoV-2, the etiologic agent of coronavirus disease 19 (COVID-19), was discovered in Wuhan (China) in December 2019 and became pandemic within a couple of months[1]. Most patients with SARS-CoV-2 infection are asymptomatic or show only mild symptoms, while some patients develop bilateral interstitial pneumonia and require oxygen support by nasal cannula or mechanical ventilation on an intensive care unit (ICU)[2].

Several risk factors for COVID-19 associated death have been identified including age, gender, adipositas, pulmonary diseases, and diabetes[3–6]. Acute respiratory distress syndrome (ARDS) and fatal outcomes are also associated with hyperinflammation and microthrombosis[7]. Activated monocytes/macrophages contribute to death in a mouse model of SARS-CoV infection[8] and also seem to play an important role in hyperinflammation in human SARS-CoV-2 infection[9,10]. Monocytes are activated by a wide range of signals, including pathogen-derived signals (e.g. nucleic acids) and T-cell-derived signals (e.g. cytokines)[11]. The innate immune system seems to be insufficient to rapidly clear SARS-CoV-2 infection by itself, as SARS-CoV-2-specific T- and B-cell responses develop in most patients[12–14]. Virus-specific T-cell responses and neutralizing antibodies, typically generated in a T-cell-dependent manner, are considered essential for viral clearance[15]. Potential benefits of plasma obtained from convalescent COVID-19 patients also points into this direction[16,17]. On the other hand, hyperinflammation in COVID-19 patients might be caused by excessive T-cell responses that activate monocytes and neutrophils. Beneficial effects of dexamethasone indicate that hyperinflammation contributes to fatal outcomes[18,19]. However, dexamethasone does not only interfere with T-cell activation and could also exert therapeutic effects by blocking excessive innate immune responses against viral components.

Current data on polyclonal T-cell reactivity in COVID-19 are conflicting and indicate that patients with severe COVID-19 can have either insufficient[20,21] or excessive[22–24] T-cell responses, as recently summarized[25]. T-cell responses were measured by intracellular cytokine staining of PBMCs, cultured with phorbolesters and ionomycin, by quantification of cytokines after polyclonal stimulation, or by expression of surface markers on T cells. IFN-gamma expression for example was found to be unchanged[26], increased[23,24], or decreased[20,21]. In contrast, SARS-CoV-2-specific T-cell responses were repeatedly reported to be reduced in critically ill and older COVID-19 patients and associated with an unfavorable outcome[27,28].

We applied a sensitive method to analyze polyclonal T-cell reactivity in COVID-19 patients and controls. T-cell activation was measured by downstream effects on responder cells, such as basophils, pDCs, monocytes, and neutrophils in whole blood. These assays turned out to be much more sensitive and consistent than classical readouts with PBMCs. We found that impaired T-cell reactivity is already evident in mild disease, caused by plasma components present in COVID-19 patients and, strongly associated with disease severity and prolonged viral replication. Gender-specific differences in T-cell-induced downstream effects on monocytes were observed. Finally, we developed a score to predict the fatal outcome to identify patients that may benefit from strategies to overcome T-cell hyporeactivity.

## Results
**Clinical characterization of COVID-19 patients**. We analyzed 55 patients with COVID-19 infection admitted to the University Hospital Regensburg from March to July 2020. SARS-CoV-2-infection was diagnosed by RT-PCR from oropharyngeal swabs. Patients were stratified into "non-ventilated" and "ventilated" patients according to the need for mechanical ventilation. Ventilated patients were further subgrouped into "survived", if they could be discharged from the ICU, or into "dead" if they died in the ICU. In addition, 42 healthy controls were included in the study. Clinical characteristics of patients and controls are shown in Table 1. In most patients, several consecutive blood samples were available resulting in a total of 188 samples. In total, 14 patients were first sampled on mechanical ventilation and also after weaning from ventilation. Samples of these patients were assigned into the subgroup "ventilated" or "non-ventilated" according to the ventilation status at the time of sampling (Supplementary Table 1).

In accordance with published data[3,4], male gender was more frequent in ventilated than non-ventilated patients (80 versus 52%). Ventilated patients also had a highly significant longer period of virus replication (28 days versus 10 days) that was defined as the period from the day of first clinical symptoms to the day of the last positive virus RT-PCR. Comorbidities were similar in both groups, while bacterial superinfection, thromboembolic complications, and treatment with antibiotics or convalescent plasma were more frequent in ventilated patients (Table 1). As described before[29], ventilated patients displayed higher plasma concentrations for biomarkers indicating inflammation (CRP, IL-6), liver and muscle dysfunction (LDH, Bilirubin, CK), and intravascular coagulation (D-dimer). Ventilated "dead" patients showed higher leukocyte counts and higher biomarkers for inflammation and liver dysfunction than ventilated "survived" patients (Supplementary Table 1).

**Impaired polyclonal T-cell reactivity is a hallmark of COVID-19 patients and correlates with disease severity**. To measure T-cell reactivity, we used fresh heparinized whole blood and polyclonally activated the T cells with immobilized anti-CD3 antibodies. T-cell activation was quantified not only by cytokine release but also by downstream effects on various cell types including basophils, pDCs, monocytes, and neutrophils. In preparation for the study, we have established that T-cell activation results in strong activation of basophils measured by upregulation of CD203c and downregulation of the IL-3-receptor beta-chain CD131 (used in this study). Basophils strongly express IL-3-receptors but only low levels of GM-CSF- and IL-5-receptors that also use CD131 for signal transduction[30]. Thus, among a variety of cytokines IL-3 is mainly responsible for downregulation of CD131 on basophils (Supplementary Fig. 1a). pDCs express IL-3- and GM-CSF-receptors; thus both cytokines contribute to downregulation of CD131 on pDCs (Supplementary Fig. 1b). CD14 + monocytes respond to T-cell activation by a very pronounced upregulation of the IL-3-receptor alpha-chain CD123. Various cytokines, including IL-3, GM-CSF, IL-2, IL-15, IFN-gamma, IFN-alpha, TNF-alpha induce a strong upregulation of CD123 on monocytes (Supplementary Fig. 1c). Thus monocytes integrate a larger set of direct and indirect T-cell-derived signals. Similarly, neutrophils respond to a large set of cytokines with upregulation of CD11b (Supplementary Fig. 1d).

T-cell activation with immobilized anti-CD3 in whole blood of healthy controls resulted in a pronounced downregulation of CD131 on basophils (to 13%) (Fig. 1a) as well as upregulation of CD123 on monocytes (to 1225%) (Fig. 1c) and CD11b on neutrophils (to 1232%) (Fig. 1e). In COVID-19 patients these changes were much weaker. Ventilated COVID-19 patients showed no significant downregulation of CD131 on pDCs, a weak downregulation of CD131 on basophils (to 77%), a weak upregulation of CD123 on monocytes (to 345%), and of CD11b on neutrophils (to 424%). Responses of non-ventilated patients were in-between the responses of ventilated patients and healthy

**Table 1 Demographics and clinical information from study patients and healthy controls.**

| | Healthy | COVID-19 all | COVID-19 non-ventilated | COVID-19 ventilated all | COVID-19 ventilated survived | COVID-19 ventilated dead |
|---|---|---|---|---|---|---|
| Number of patients (n) | 42 | 55 | 25 | 30 | 23 | 7 |
| *Demographics* | | | | | | |
| Mean age (years, min-max) | 37.8[a] (19–78) | 56.8 (19–78) | 55.1 (19–78) | 58.3 (31–73) | 57.7 (31–73) | 60.1 (51–68) |
| Sex (% male) | 15 (36%)[b] | 37 (67%) | 13 (52%) | 24 (80%)[c] | 17 (74%) | 7 (100%) |
| *Virus persistence* | | | | | | |
| Mean, days | | 20 | 10 | 28[d] | 29 | 23 |
| *Comorbidities, n (%)* | | | | | | |
| Smoking | | 5 (9%) | 4 (16%) | 1 (3%) | 1 (4%) | 0 |
| Hypertension | | 28 (51%) | 11 (44%) | 17 (57%) | 12 (52%) | 5 (71%) |
| Diabetes mellitus | | 15 (27%) | 6 (24%) | 9 (30%) | 6 (26%) | 3 (43%) |
| Cardiovascular disease | | 5 (9%) | 4 (16%) | 1 (3%) | 1 (4%) | 0 |
| Lung disease | | 12 (22%) | 6 (24%) | 6 (20%) | 3 (13%) | 3 (43%) |
| Immunosuppression | | 7 (13%) | 5 (20%) | 2 (7%) | 1 (4%) | 1 (14%) |
| *Intercurrent events, n (%)* | | | | | | |
| Bact. superinfection | | 31 (56%) | 5 (20%) | 26 (87%)[e] | 20 (87%) | 6 (86%) |
| ECMO | | 10 (18%) | 0 | 10 (33%)[f] | 8 (35%) | 2 (29%) |
| Thromboembolic complications | | 14 (25%) | 1 (4%) | 13 (43%)[g] | 11 (48%) | 2 (29%) |
| *Treatment, n (%)* | | | | | | |
| Hydroxychloroquine | | 2 (4%) | 0 | 2 (7%) | 2 (9%) | 0 |
| Lopinavir/ritonavir | | 1 (2%) | 0 | 1 (3%) | 1 (4%) | 0 |
| Convalescent plasma | | 23 (42%) | 5 (20%) | 18 (60%)[h] | 12 (52%) | 6 (86%) |
| Glucocorticoids | | 17 (31%) | 5 (20%) | 12 (40%) | 9 (39%) | 3 (43%) |
| Antibiotics | | 38 (69%) | 9 (36%) | 29 (97%)[i] | 23 (100%) | 6 (86%) |

COVID-19 patients were subgrouped into non-ventilated and ventilated patients. Non-ventilated patients were hospitalized on a normal ward and received oxygen by nasal cannula as needed. Ventilated patients were treated in an ICU by mechanical ventilation. Ventilated patients were further subgrouped into patients that were discharged from the ICU ("Survived") and patients that died in the ICU ("Dead"). Virus persistence was calculated from the day of the first clinical symptoms to the day of the last positive virus PCR. Two-tailed unpaired t test was used to calculate statistical differences between non-ventilated and ventilated patients as well as between "Survived" and "Dead" patients.
[a]$P < 0.0001$.
[b]$P = 0.0018$.
[c]$P = 0.0276$.
[d]$P < 0.0001$.
[e]$P < 0.0001$.
[f]$P = 0.0010$.
[g]$P = 0.0006$.
[h]$P = 0.0022$.
[i]$P < 0.0001$.

controls. Duration of T-cell hyporeactivity as defined by weak upregulation of CD123 and CD11b was significantly associated with prolonged viral replication in these patients (Supplementary Table 2). Absolute expression levels of surface markers on basophils, pDCs, monocytes, and neutrophils and representative FACS dot plots of unstimulated and anti-CD3-stimulated whole blood cells are shown in Supplementary Fig. 1 and Supplementary Fig. 3. Downregulation of CD131 on basophils was almost completely dependent on the release of IL-3 as shown with a blocking antibody against IL-3, while downregulation of CD131 on pDCs was only partially dependent on IL-3 (Supplementary Fig. 2a, b). Inhibition of IL-3 did not reduce the upregulation of CD123 on monocytes or CD11b on neutrophils (Supplementary Fig. 2c, d), most likely due to the high redundancy of T-cell-derived signals with effects on monocytes and neutrophils (Supplementary Fig. 1). Stratification of ventilated COVID-19 patients according to their clinical outcome (survival vs. death) revealed that downregulation of CD131 on pDCs or basophils was low in both groups (Fig. 1b). In contrast, upregulation of CD123 on monocytes and CD11b on neutrophils, that integrate a larger set of signals from activated T cells, were almost absent in COVID-19 patients with fatal outcome, but still detectable in ventilated patients who recovered and were later discharged from the ICU (Fig. 1d, f). In healthy males and non-ventilated male COVID-19 patients, T-cell activation resulted in a significantly stronger upregulation of CD123 on monocytes than in females, suggesting a stronger T-cell–monocyte connection in males (Supplementary Fig. 4c). In ventilated COVID-19 patients, the impairment of T-cell-induced CD123 upregulation on monocytes was very pronounced in both genders (Supplementary Fig. 4c). Basophils, pDCs, and neutrophils responded equally to T-cell activation in males and females (Supplementary Fig. 4a, b, d).

To exclude that downstream responder cells like basophils, monocytes, and neutrophils are anergic to cytokine stimulation in COVID-19 patients, we incubated whole-blood samples of 10 healthy controls, 13 non-ventilated, and 15 ventilated patients with anti-CD3 or just with recombinant IL-3 and GM-CSF for 24 h (Supplementary Fig. 5). While the response of basophils, monocytes, and neutrophils to T-cell activation with anti-CD3 was very low in COVID-19 patients, their response to recombinant IL-3 and GM-CSF was fully preserved in a cell-type-specific manner (preferential response of basophils to IL-3, of monocytes to IL-3 and GM-CSF, and of neutrophils to GM-CSF).

CD4 + and CD8 + T-cell counts in whole-blood samples were similar in ventilated patients, non-ventilated patients, and healthy controls, arguing against the hypothesis that only differences in T-cell numbers account for the differences observed in whole blood after T-cell activation (Fig. 2a, b). Only in ventilated patients with fatal outcome CD4 + and CD8 + T-cell counts were reduced.

We also measured selected T-cell-derived cytokines by ELISA in the supernatant of T-cell-activated whole blood (Fig. 2c–f). We found no significant anti-CD3-induced release of IL-3, GM-CSF, and IFN-gamma in ventilated COVID-19 patients, but a considerable release in non-ventilated patients and healthy

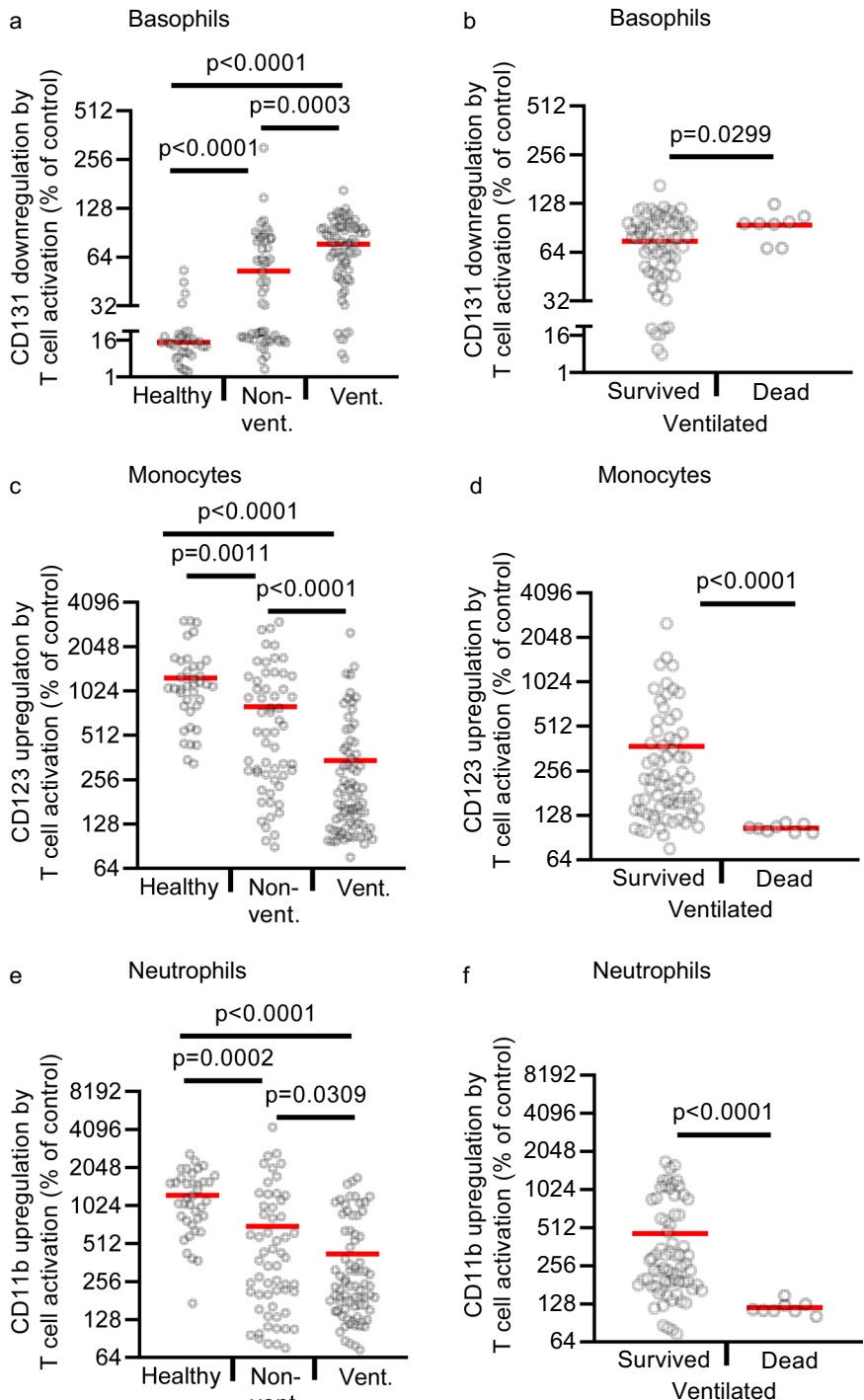

**Fig. 1 T-cell reactivity in COVID-19 patients and healthy controls. a–f** Whole-blood samples from 38 healthy controls (healthy; $n = 38$ biologically independent samples), 33 non-ventilated (non-vent.; $n = 58$ biologically independent samples), and 21 mechanically ventilated (vent.; $n = 77$ biologically independent samples) COVID-19 patients were cultured with or without immobilized anti-CD3 for 24 h. Ventilated patients were stratified into "survived" (17 patients, $n = 69$ biologically independent samples) and "dead" (4 patients, $n = 8$ biologically independent samples). Expression of indicated surface markers was quantified by flow cytometry on basophils (**a**, **b**), CD14 + monocytes (**c**, **d**), and neutrophils (**e**, **f**). Values depict the ratio of surface marker expression with anti-CD3 to surface marker expression without anti-CD3. Each sample is represented by one dot, and the mean is marked in red. One-way ANOVA with Bonferroni multiple comparison test was used for **a**, **c**, and **e**. Two-tailed unpaired $t$ test with Welch´s correction was used for **b**, **d**, and **f**. Source data are provided as a Source Data file.

controls. These data are largely consistent with the above described downstream effects of T-cell activation on basophils, pDCs, monocytes, and neutrophils. However, the cytokine levels are more variable and show no significant differences between non-ventilated patients and healthy controls. Thus, downstream cellular

effects of T-cell activation are more sensitive than measurement of single cytokines by ELISA. Although some COVID-19 patients showed a high release of IL-10 in the supernatant of T-cell-activated whole blood, the difference between healthy controls and patients was not significant. The almost complete inhibition of

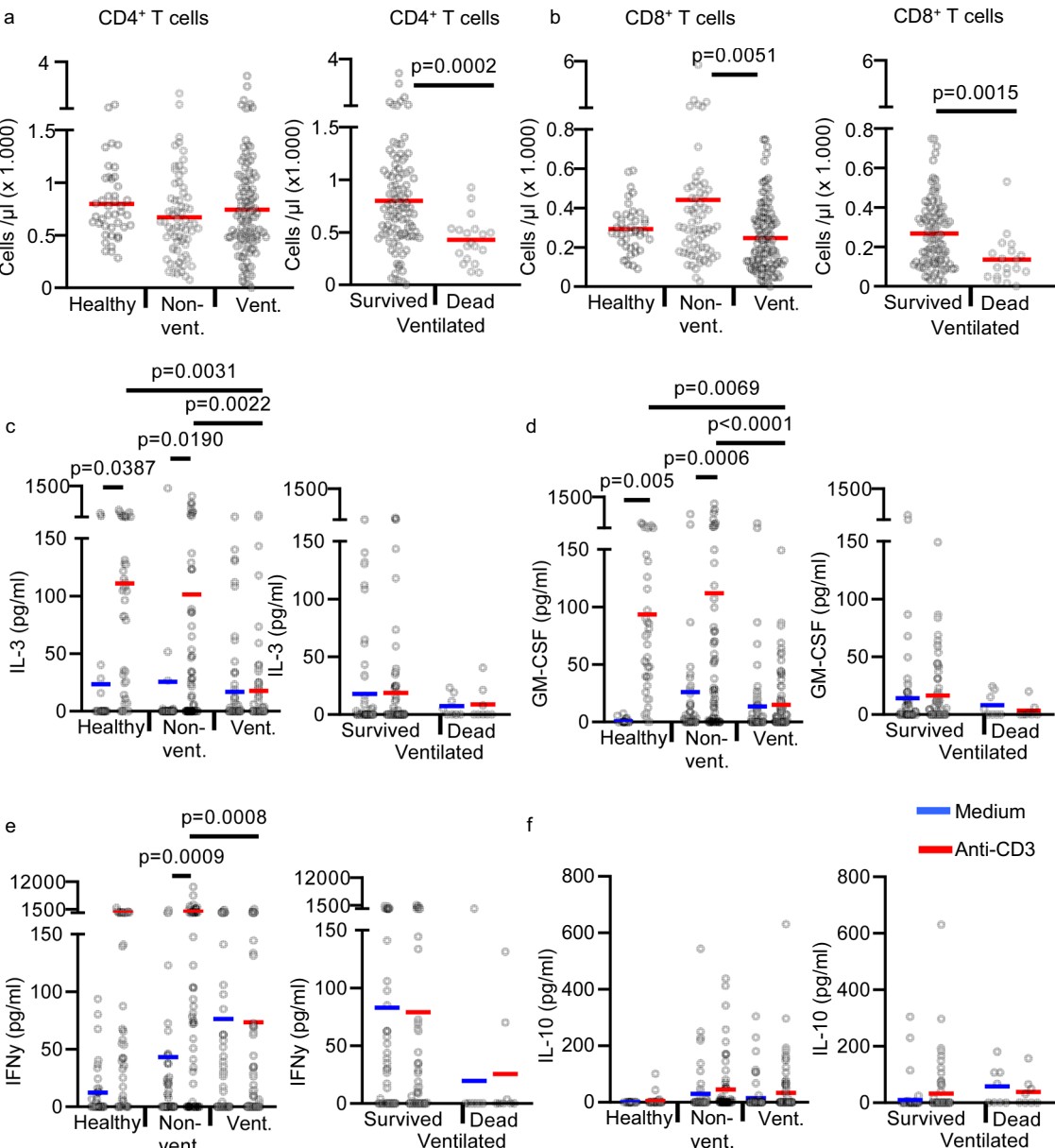

**Fig. 2 T-cell counts and cytokine release in anti-CD3 activated whole-blood samples. a, b** Absolute CD4 + and CD8 + T-cell counts in non-cultured fresh blood of 42 healthy controls (healthy; $n = 42$ biologically independent samples), 39 non-ventilated COVID-19 patients (non-vent.; $n = 68$ biologically independent samples) and 30 ventilated COVID-19 patients (vent.; $n = 120$ biologically independent samples). Ventilated patients were stratified into "survived" (23 patients, $n = 101$ biologically independent samples) and "dead" (7 patients, $n = 19$ biologically independent samples). Each sample is represented by one dot, and the mean is marked in red. **c-e** Whole blood from 38 healthy controls (healthy; $n = 38$ biologically independent samples), 33 non-ventilated (non-vent.; $n = 58$ biologically independent samples) and 21 mechanically ventilated (vent.; $n = 77$ biologically independent samples) COVID-19 patients was cultured with or without immobilized anti-CD3 for 24 h. Ventilated patients were stratified into "survived" (17 patients, $n = 69$ biologically independent samples) and "dead" (4 patients, $n = 8$ biologically independent samples). Concentrations of IL-3 (**c**), GM-CSF (**d**), IFN-γ (**e**), and IL-10 (**f**) were measured in the culture supernatant by ELISA. Each sample is represented by one dot, and the mean is marked in blue (medium) or red (anti-CD3). One-way ANOVA with Bonferroni multiple comparison test was used. Two-tailed unpaired $t$ test was used for the analysis of "Survived" and "Dead". Source data are provided as a Source Data file.

cytokine release from activated T cells in ventilated COVID-19 patients strongly suggests that both CD4 + and CD8 + T cells are hyporeactive, as IL-3 and IFN-gamma are released from both, CD4 + and CD8 + T cells.

In order to find out whether T-cell hyporeactivity in COVID-19 patients is T-cell intrinsic or extrinsic, we exposed washed whole blood from a healthy donor with plasma from 10 healthy donors, 13 non-ventilated, and 19 ventilated COVID-19 patients. Plasma from non-ventilated patients and even more ventilated patients markedly

suppressed T-cell activation, as shown by a much lower down-regulation of CD131 on basophils and a much lower upregulation of CD123 on monocytes and CD11b on neutrophils (Fig. 3). The suppressive effects of COVID-19 plasma could not be explained by immunosuppressive medication, as only 23% of the non-ventilated and 33% of the ventilated patients received steroids. No other immunosuppressive agents were given to any of the patients. In Fig. 3, the plasma samples of patients with steroids are marked in blue and show no increased T-cell suppression.

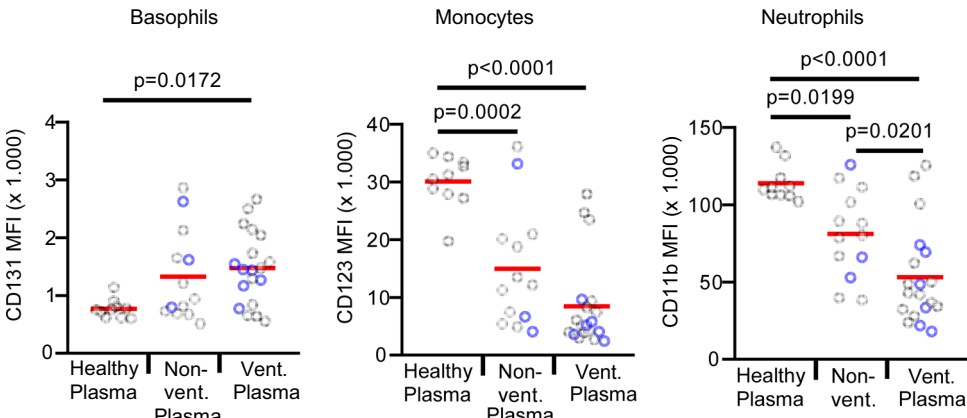

**Fig. 3 Plasma from COVID-19 patients has pronounced T-cell suppressive activity.** Whole blood from a healthy donor was washed twice with medium to remove the plasma. Plasma from 10 healthy controls (healthy plasma; $n = 10$ biologically independent samples), 13 non-ventilated (non-vent. plasma; $n = 13$ biologically independent samples), and 15 mechanically ventilated (vent. plasma; $n = 19$ biologically independent samples) COVID-19 patients was added and samples were cultured with anti-CD3 for 24 h. In all, 23% of the non-ventilated and 33% of the ventilated COVID-19 patients were treated with steroids (marked in blue). None of the patients was treated with other immunosuppressive agents. Expression of indicated surface markers was quantified by flow cytometry on basophils, CD14 + monocytes and neutrophils. The absolute expression values of indicated markers are shown as mean fluorescence intensity (MFI) on basophils, CD14 + monocytes, and neutrophils. Each sample is represented by one dot, and the mean is marked in red. One-way ANOVA with Bonferroni multiple comparison test was used. Source data are provided as a Source Data file.

We tested several approaches to overcome T-cell hyporeactivity in COVID-19 patients. The addition of recombinant IL-2 markedly improved T-cell reactivity in non-ventilated patients but was less effective in ventilated patients (Supplementary Fig. 6). Blockade of IL-10 with a blocking antibody and addition of excess L-tryptophan to overcome potentially increased tryptophan-degradation by indoleamine 2,3-dioxygenase (IDO) had no beneficial effects on T-cell reactivity (Supplementary Fig. 7).

**PBMCs are not suitable to quantify polyclonal T-cell hyporeactivity in COVID-19 patients.** We also used purified PBMCs to polyclonally activate the T cells with soluble anti-CD3 for 24 h. Downstream effects of T-cell activation on pDCs and neutrophils could not be analyzed in PBMC samples, because neutrophils are absent and pDC numbers were too low. Basophils and monocytes responded only weakly to T-cell-derived signals without significant differences between non-ventilated and ventilated patients (Supplementary Fig. 8a, b). Upregulation of CD25 on T cells (Supplementary Fig. 8c, d) and release of T-cell-derived cytokines (Supplementary Fig. 8e) were clearly induced by anti-CD3; however, only for GM-CSF a significant difference was detectable between non-ventilated and ventilated patients.

**T-cell hyporeactivity is reversible in critically ill COVID-19 patients.** In several COVID-19 patients, multiple blood samples were available over a prolonged period of time (Supplementary Fig. 9). Four cases of ventilated patients that could finally be weaned from mechanical ventilation and two cases of patients that died in the ICU are shown. Patients with fatal outcomes showed an almost complete T-cell hyporeactivity in multiple samples over up to 14 days. In contrast, T-cell hyporeactivity was less pronounced in patients with favorable outcomes. In general, their T-cell reactivity appeared to improve already before weaning and was improved after successful weaning. This shows on an individual basis that T-cell hyporeactivity is reversible and correlates to disease severity.

**Immunophenotyping of fresh blood samples in COVID-19 patients.** We performed immunophenotyping of fresh blood samples of COVID-19 patients and healthy controls with a focus on innate cells like basophils, pDCs, monocytes, and neutrophils (Fig. 4). The gating strategy is shown in Supplementary Fig. 10. Basophil counts were markedly reduced in ventilated COVID-19 patients that subsequently died in the ICU (Fig. 4a). Consistent with published data, we also found decreased numbers of pDCs in ventilated COVID-19 patients (Fig. 4b).

Total neutrophil counts were higher in ventilated COVID-19 patients and also higher in ventilated patients with a fatal outcome (Fig. 4c). Expression of CD11b on neutrophils showed the same pattern as CD169 on monocyte. CD11b was significantly higher in non-ventilated COVID-19 patients and returned almost to baseline in ventilated and deceased patients (Fig. 4d).

CD16 + monocytes but not the classical CD14 + monocytes were decreased in non-ventilated and ventilated patients (Fig. 4e, g). Both monocyte populations showed significantly increased expression of CD169 (Siglec-1) in non-ventilated patients compared to healthy controls (Fig. 4f, h). CD169 is one of the most prominent type I IFN-regulated genes on human monocytes[31]. In ventilated COVID-19 patients, expression of CD169 returned almost to baseline, suggesting an impaired type I interferon production in critically ill COVID-19 patients. Reduced type I interferon responses were recently discovered by transcriptional profiling in critically ill COVID-19 patients[32].

**Design and assessment of a predictive score.** The immunological characteristics of COVID-19 patients lead us to develop predictive scores for the survival of ventilated patients. We have incorporated three parameters that discriminated best between various groups of COVID-19 patients and calculated predictive values for the death of ventilated patients (Table 2). The first two parameters are CD123 upregulation on monocytes and CD11b-upregulation on neutrophils induced by anti-CD3 in whole blood with a cutoff of 130%. Both parameters reflect the degree of T-cell hyporeactivity; however, they are not interchangeable, as monocytes and neutrophils sense different sets of T-cell-derived signals. The third parameter is the basophil count in the peripheral blood with a cutoff of 25/µl. All three parameters should be easy to establish in different labs because the counting of basophils is routine and no standardized absolute expression levels are required for upregulation of CD123 and CD11b as only relative changes are calculated. By combination of all three

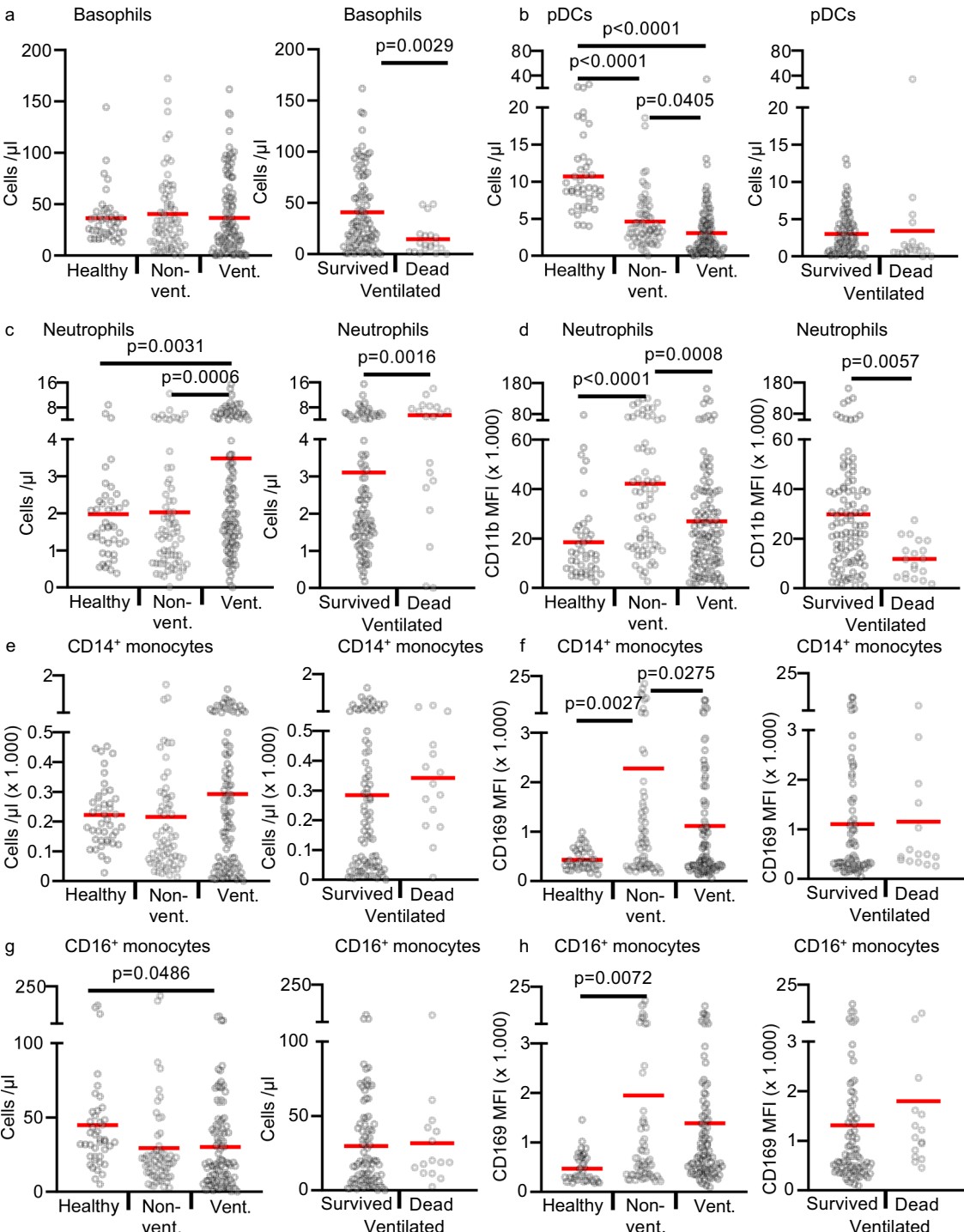

**Fig. 4 Immunophenotyping of fresh peripheral blood cells in COVID-19 patients and healthy controls. a–d** 42 healthy controls (healthy; $n = 42$ biologically independent samples), 39 non-ventilated (non-vent.; $n = 68$ biologically independent samples), and 30 mechanically ventilated (vent.; $n = 120$ biologically independent samples) COVID-19 patients were analyzed. Ventilated patients were stratified into "survived" (23 patients, $n = 101$ biologically independent samples) and "dead" (7 patients, $n = 19$ biologically independent samples). **a, b** Absolute basophil and pDC counts. **c, d** Absolute neutrophils counts and their expression of CD11b given as mean fluorescence intensity (MFI). **e–h** 42 healthy controls (healthy; $n = 42$ biologically independent samples), 38 non-ventilated (non-vent.; $n = 62$ biologically independent samples), and 30 mechanically ventilated (vent.; $n = 102$ biologically independent samples) COVID-19 patients were analyzed. Ventilated patients were stratified into "survived" (23 patients, $n = 87$ biologically independent samples) and "dead" (7 patients, $n = 15$ biologically independent samples). The number of samples is lower than in **a–d** because the antibodies for this panel were temporarily not available by suppliers. **e, f** Absolute CD14 + + CD16- monocyte counts and their surface expression of CD169. **g, h** Absolute CD16 + monocyte counts and their surface expression of CD169. Each sample is represented by one dot, and the mean is marked in red. One-way ANOVA with Bonferroni multiple comparison test was used for the analysis of "healthy", "non-vent", and "vent". Two-tailed unpaired *t* test was used for the analysis of "survived" and "dead". Source data are provided as a Source Data file.

**Table 2 Predictive score for death in ventilated COVID-19 patients.**

**Stimulated blood:**

| 77 complete datasets of ventilated patients | Right positive (of 8 dead) | False positive (of 69 survived) | Sens. | Spe. | NPV | PPV |
|---|---|---|---|---|---|---|
| Weak upregulation of CD123 on Monos + weak upregulation of CD11b on Neutros + low Basophil count | 6 | 0 | 75% | 100% | 94% | 100% |
| Weak upregulation of CD11b on Neutros + low Basophil count | 6 | 1 | 75% | 99% | 94% | 93% |
| Weak upregulation of CD123 on Monos + weak upregulation of CD11b on Neutros | 7 | 3 | 88% | 96% | 97% | 83% |
| Weak upregulation of CD123 on Monos + low Baso count | 7 | 6 | 88% | 92% | 97% | 71% |
| Weak upregulaiton of CD11b on Neutros | 7 | 7 | 88% | 90% | 97% | 67% |
| Weak upregulation of CD123 on Monos | 8 | 15 | 100% | 80% | 100% | 52% |
| Low Baso count | 7 | 32 | 88% | 54% | 95% | 31% |

Three parameters were used individually or in combination to predict fatal outcome in ventilated COVID-19 patients. Absolute basophil counts in fresh peripheral blood. Basophil counts <25/µl defined "Low Baso count". Upregulation of CD123 on CD14 + monocytes and CD11b on neutrophils defined as the ratio of surface marker expression with anti-CD3 to surface marker expression without anti-CD3. Upregulation of <130% defined "Weak upregulation". In the combined scores, a logical AND combination was used that required all parameters to be fulfilled. For all three parameters, 77 datasets from 21 ventilated COVID-19 patients ($n = 77$ biologically independent samples) were available. In all, 17 ventilated patients were discharged from the ICU (survived) ($n = 69$ biologically independent samples) and 4 patients died on the ICU ($n = 8$ biologically independent samples). The number of right and false positive samples, the test sensitivity (Sens.), specificity (Spe.), negative (NPV), and positive predictive value (PPV) for predicting death are shown.

parameters, a positive predictive value of 100% and a negative predictive value of 94% were reached for fatal outcomes of patients on mechanical ventilation.

## Discussion

We have shown that low polyclonal T-cell reactivity is a hallmark of COVID-19 patients and strongly correlated with disease severity. T-cell reactivity was much lower in ventilated than in non-ventilated hospitalized patients, which was again somewhat lower than in healthy controls. T-cell hyporeactivity was only detectable in assays with whole blood but not with purified PBMC and could be attributed to suppressive components present in the plasma of COVID-19 patients. Incubation of whole blood from a healthy donor with plasma samples from ventilated COVID-19 patients resulted in a pronounced polyclonal T-cell hyporeactivity. Neither immunosuppressive drugs potentially present in the plasma of COVID-19 patients nor IL-10 or degradation of L-tryptophan could explain the inhibitory effects of COVID-19 plasma. With PBMCs and classical methods to study T-cell activation conflicting data have been published showing unaltered, increased, or decreased polyclonal T-cell responses in critically ill COVID-19 patients[20,21,23–26]. Recently, impaired SARS-Cov-2-specific CD4 + and CD8 + T-cell responses were found to be associated with older age and more severe outcome, while humoral immune responses were not predictive[27,28]. Our data are in line with these findings and additionally suggest that critically ill COVID-19 have a more general T-cell suppression that could also impair the development of SARS-CoV-2-specific T-cell responses. Many critically ill COVID-19 patients suffer from fungal infections, which may be caused by the general T-cell suppression in these patients[33,34].

Measurement of T-cell activation by downstream effects on responder cells was much more sensitive and suitable to detect impairment of T-cell activation and differences between groups of patients than measurement of T-cell-derived cytokines in the supernatant. The responder cells integrate a large set of signals from activated T cells in a very sensitive and cell-type-specific manner.

Our data indicate that low T-cell reactivity rather than anergy of downstream responder cells accounts for the low response of basophils, pDCs, monocytes, and neutrophils, because relevant T-cell-derived cytokines like IL-3, GM-CSF, and IFN-gamma were reduced in a similar manner and neutralization of IL-3 markedly blocked the downstream effects on basophils. In addition, responder cells in patient samples were fully responsive towards

stimulation with recombinant IL-3 and GM-CSF, further arguing against the energy of responder cells.

The severe T-cell hyporeactivity in critically ill COVID-19 patients does not seem to be a mere consequence of being critically ill, because T-cell hyporeactivity was already seen in non-ventilated COVID-19 patients. The appearance of T-cell hyporeactivity could be a counter-regulatory mechanism to the hyperinflammation present in COVID-19 patients or a strategy of the virus to interfere with specific host immune responses and may explain the prolonged viral replication seen in the group of ventilated patients. Prolonged viral replication in turn may result in more tissue destruction, more activation of innate immune cells by viral components and more inflammation. The pronounced T-cell hyporeactivity in critically ill patients with hyperinflammation strongly argues against a major direct role of T cells for hyperinflammation. Immunosuppressive approaches targeting T cells are unlikely to improve hyperinflammation, while strategies to overcome the pronounced T-cell hyporeactivity in critically ill COVID-19 patients could be a promising therapeutic approach to improve virus clearance without aggravating the hyperinflammation.

Immunophenotyping revealed low counts of basophils and pDCs and a low expression of CD11b on neutrophils in severely ill COVID-19 patients. It is known that IL-3 is the major cytokine to increase basophil counts[30]. IL-3 and/or GM-CSF are also important survival factors for basophils and pDCs[35,36]. Thus the lower counts of pDCs and basophils in critically ill patients may reflect not only their increased recruitment into tissues but also reduced survival and production of cells due to diminished secretion of IL-3 and GM-CSF by anergic T cells. Compared to healthy controls, basal CD11b expression on neutrophils is higher in non-ventilated and somewhat higher in ventilated patients. This indicates that COVID-19 leads to an activation of neutrophils. The pronounced T-cell hyporeactivity in ventilated patients may explain why basal CD11b expression is significantly lower in ventilated compared to non-ventilated patients and again lower in patients with a fatal outcome.

Gender-specific analysis showed that the T-cell-induced upregulation of CD123 on monocytes was significantly higher in healthy males and non-ventilated male patients than in the same groups of females. As IL-3 contributes to the activation of monocytes[37], the stronger T-cell–monocyte link in males may explain the recently described higher plasma levels of IL-8 and IL-18 in male COVID-19 patients[38].

We have developed a score based on downstream effects of T-cell activation on monocytes and neutrophils, and absolute basophil counts to predict fatal outcome in ventilated COVID-19 patients. We would like to point out, that the score must be confirmed in additional cohorts of COVID-19 patients. The scores may be clinically useful to identify patients that may benefit from strategies to overcome T-cell hyporeactivity.

Collectively our data suggest that approaches to overcome T-cell hyporeactivity combined with concepts to specifically suppress hyperinflammation induced by innate immune cells, particularly monocytes/macrophages could be promising immunotherapy for critically ill COVID-19 patients.

## Methods

**Study subjects and sampling.** A total of 55 adult patients diagnosed with COVID-19 after testing positive for SARS-CoV-2 RNA by qPCR and 42 adult healthy volunteers without any clinical signs of COVID-19 infection were enrolled from March to July 2020 at the University Hospital Regensburg. This study was approved by the Research Ethics Committee from the University Hospital Regensburg (Study and Approval Number: 20-1785-101). The study was performed in accordance with all relevant ethical regulations for work with human participants. Informed consent and consent to publish were obtained from all enrolled patients (or their legal representatives) and healthy volunteers.

The cohort of COVID-19 patients was stratified into non-ventilated ($n = 25$) and mechanically ventilated ($n = 30$). Ventilated patients were further subgrouped according to their outcome (discharged from the ICU = "survived", or death on the ICU = "dead") (Table 1). In most patients, several consecutive blood samples were available resulting in a total of 188 samples. In total, 14 patients were first sampled on mechanical ventilation and also after discharge from the ICU. Samples of these patients were assigned into the subgroup "ventilated" or "non-ventilated" according to the ventilation status at the time of sampling (Supplementary Table 1). None of the patients received prednisolone equivalents ≥40 mg/day for treatment of COVID-19.

Fresh whole-blood samples for immediate immunophenotyping by flow cytometry with at least one antibody panel were available from all participants. Sufficient amounts of fresh blood to also perform whole-blood stimulation were available from 38 healthy controls (38 samples), 33 non-ventilated COVID-19 patients (58 samples), and 21 mechanically ventilated COVID-19 patients (77 samples). For subgroup analysis the mechanically ventilated patients were again stratified according to their outcome into the groups "survived" (17 patients, 69 samples) and "dead" (4 patients, 8 samples). Peripheral blood mononuclear cells (PBMCs) were prepared from 25 non-ventilated patients (36 samples) and 16 ventilated COVID-19 patients (42 samples). For subgroup analysis, the mechanically ventilated patients were again stratified according to their outcome into the groups "survived" ($n = 14$, 39 samples) and "dead" ($n = 2$, 3 samples). The exact numbers of patients and samples for each readout are provided in the figure legends, as for some readouts, not enough material was available.

**Cell culture media, cytokines, and stimulants.** For PBMC cultures, we used RPMI 1640 medium (Gibco, Karlsruhe, Germany) containing 1% penicillin/streptomycin (Gibco, Karlsruhe, Germany) and 1% L-glutamine (Gibco, Karlsruhe, Germany). For whole-blood stimulations, we used pure RPMI 1640 medium (Gibco, Karlsruhe, Germany). Anti-CD3 (clone OKT3, eBiosciences, San Diego, CA, Cat. # 16-0037-85) was used at 5 µg/ml. The blocking anti-IL-3 antibody (clone P8C11) was generated in our lab and used at a concentration of 10 µg/ml. The blocking anti-IL-10 antibody (clone JES3-19F1, BioLegend, San Diego, USA,, Cat. # 506813) was used at 20 µg/ml. IL-3 (BioLegend) and GM-CSF, IL-2, IL-4, IL-5, IL-6, IL-15, IFN-γ, IFN-α, TNF-α (Peprotech, Cranbury, USA) were used at 20 ng/ml. L-tryptophan (Sigma-Aldrich, Darmstadt, Germany) was used at 100 µg/ml.

**Whole-blood stimulation.** For T-cell activation, assays tubes (BD Bioscience) were precoated with 300 µl anti-CD3 antibodies (5 µg/ml in PBS) at 37 °C for 4 h and washed twice with PBS. In all, 100 µl of heparinized blood were diluted with 200 µl RPMI 1640 medium and added to the precoated or to uncoated tubes and incubated at 37 °C in 5% $CO_2$ for 24 h. Where indicated a blocking anti-IL-3 antibody (10 µg/ml), a blocking anti-IL-10 antibody (20 µg/ml), IL-2 (20 ng/ml), GM-CSF (10 ng/ml), or IL-3 (10 ng/ml) were added to the samples. The supernatants were recovered and released cytokines analyzed by ELISA. Cells were analyzed by multi-parametric flow cytometry. For cytokine stimulation, 100 µl whole blood was diluted with 200 µl RPMI 1640 medium and added to uncoated tubes (BD Bioscience) together with 20 ng/ml of cytokines (IL-2, IL-3, IL-4, IL-5, IL-6, IL-15, GM-CSF, IFN-γ, IFN-α, and TNF-α) at 37 °C in 5% $CO_2$ for 24 h. Cells were analyzed by flow cytometry.

**Whole-blood stimulation with the transfer of plasma.** Assay tubes (BD Bioscience) were precoated with 300 µl anti-CD3 antibodies (5 µg/ml in PBS) at 37 °C for 4 h and washed twice with PBS. Heparinized whole blood from a healthy

donor was washed twice with medium to remove the plasma. In each precoated or uncoated assay tubes, 100 µl of washed whole blood, 150 µl of the medium, and 50 µl of heparinized plasma from healthy controls, non-ventilated and ventilated COVID-19 patients was added and samples were cultured at 37 °C in 5% $CO_2$ for 24 h. Cells were analyzed by multi-parametric flow cytometry.

**Preparation and culture of PBMCs.** Peripheral blood mononuclear cells (PBMCs) were prepared by Ficoll-Paque density gradient centrifugation from heparin-anticoagulated fresh blood samples. For cryopreservation, PBMCs were resuspended in fetal calf serum (FCS) with 10% dimethylsulfoxide (DMSO) at a concentration of $1-2 \times 10^6$ cells/ml, frozen at −80 °C for 2–3 days and then transferred into liquid nitrogen. The cells were thawed in a 37 °C water bath and washed three times with medium. The viability was controlled by Trypan-blue staining and was 90–95%. PBMCs (500.000/well) were cultured in 300 µl medium/well with or without anti-CD3 (clone: OKT3) 5 µg/ml for 24 h by 37 °C and 5% $CO_2$. Thereafter supernatants were recovered and analyzed by ELISA. Cells were analyzed by flow cytometry.

**Flow cytometry.** Anticoagulated fresh blood samples (100 µl) or cells after whole-blood stimulations were incubated with various panels of the following directly labeled monoclonal antibodies for 20 min by 4 °C: anti-hCD3 APC-Cy7 (clone: SK7, BioLegend, Cat. # 344818, dilution 1:100), anti-hCD4 V500 (clone RPA-T4, BD Bioscience, Cat. # 560768, dilution 1:100), anti-hCD8 APC-eFluor 780 (clone: RPA-T8, Invitrogen, Darmstadt, Germany, Cat. # 47-0088-42, dilution 1:40), anti-hCD8 PE-Cy7 (clone: SK1, BioLegend, Cat. # 344712, dilution 1:100), anti-hCD11b PE-Cy7(clone: M1/70, BioLegend, San Diego, CA, Cat. # 101216, dilution 1:100), anti-hCD14 V500 (clone: MΦP9, BD Bioscience, Cat. # 562693, dilution 1:100), anti-hCD16 Pacific Blue (clone: 3G8, BioLegend, Cat. # 302032, dilution 1:100), anti-hCD19 Pacific Blue (clone: HIB19, BioLegend, Cat. # 302232, dilution 1:100), anti-hCD25 APC (clone: BC96, BioLegend, Cat. # 302610, dilution 1:40), anti-hCD116 FITC (clone: 4H1, BioLegend, Cat. # 305906, dilution 1:40), anti-hCD131 PE (clone: 1C1, eBioscience, Cat. # 12-1319-42, dilution 1:100), anti-hCD123 PE-Cy5 (clone: 9F5, BD Bioscience, Cat. # 551065, dilution 1:20), anti-hCD169 PE (clone: 7-239, Miltenyi Biotec, Bergisch Gladbach, Germany, Cat. # 130-098-654, dilution 1:22), anti-hCD304 APC (clone: 12C2, BioLegend, Cat. # 354506, dilution 1:40), anti-hHLA-DR II APC (clone: G-46-6, BD Bioscience, Cat. # 559866, dilution 1:100) and FITC (clone: G-46-6, BD Bioscience, Cat. # 555811, dilution 1:100). Subsequently, samples were treated with FACS Lysing Solution (BD Bioscience) for 10 min, washed with 0.9% NaCl, centrifuged, resuspended in 0.9% NaCl together with FACS counting beads (Invitrogen, Darmstadt, Germany) and acquired with a BD FACSCanto II flow cytometer and analyzed with FACSDIVA 8 software (BD Biosciences). For analysis of cultured PBMCs, no erythrocyte lysis was performed. The gating strategy for key cell populations is shown in Supplementary Fig. 10.

**Enzyme-linked immunosorbent assay (ELISA).** Concentrations of GM-CSF, IFN-gamma, and IL-10 in cell culture supernatants were measured by ELISA, according to the manufacturer's protocol (DuoSet Elisa, R&D Systems, Abington, UK). For measurement of IL-3, ELISA plates (NUNC Maxisorb, Thermofisher Scientific, Waltham, USA) were coated with a capture anti-IL-3 antibody (Clone P8C11; 5 µg/ml in PBS) in 100 µl/well at room temperature (RT) overnight. Plates were washed three times with PBS/0.05% Tween-20, blocked with PBS containing 1% bovine serum albumin (BSA) at RT for 1 h, and washed again with PBS. Samples were preincubated with 100 µg/ml mouse IgG1, kappa isotype control antibody (MOPC21, BioXCell, Lebanon, USA) for 1 h by RT and added to the plates for 2 h by RT (100 µl/well, diluted in PBS/1% BSA). Recombinant human IL-3 (BioLegend) was diluted from 7.8 to 500 pg/ml in PBS/1% BSA and served as standard. After washing, plates were incubated with 400 ng/ml HRP-labeled detection anti-IL-3 antibody (Clone 13) (100 µl/well) for 1.5 h at RT, and color reaction was performed with TMB Substrate Solution (BioLegend) according to the manufacturer's protocol.

**The predictive score for death in ventilated COVID-19 patients.** Three parameters were used individually or in combination to predict fatal outcome in ventilated COVID-19 patients. Absolute basophil counts in fresh peripheral blood. Basophil counts <25/µl defined "Low Baso count". Upregulation of CD123 on CD14 + monocytes and CD11b on neutrophils was calculated as the ratio of surface marker expression with anti-CD3 and surface marker expression without anti-CD3. Upregulation of <130% defined "Weak upregulation". In the combined scores, a logical AND combination was used that required all parameters to be fulfilled. For all three parameters, 77 datasets from 21 ventilated COVID-19 patients were available. In total, 17 ventilated patients were discharged from the ICU (survived) (69 samples) and 4 patients died on the ICU (8 samples). The number of right and false-positive samples, the test sensitivity (Sens.), specificity (Spe.) were calculated. The negative (NPV) and positive predictive value (PPV) for predicting death were calculated using a prevalence of death in ventilated COVID-19 patients of 19% (4 dead patients/21 ventilated patients).

**Statistics and reproducibility.** Statistical analyses were performed using the GraphPad Prism 8 Software. Statistical differences between more than two different cell stimulations or patient cohorts were calculated using one-way ANOVA with Bonferroni multiple comparisons, as indicated in the figures. For two-group comparisons, two-

tailed unpaired *t* test was used as indicated in the figures. In Supplementary Fig. 1, the experimental findings were successfully replicated twice. In all other cases, no replications were done, because only one sample per patient per day was available.

**Reporting summary**. Further information on experimental design is available in the Nature Research Reporting Summary linked to this paper.

## Data availability
The datasets generated during and/or analyzed during this study are available from the corresponding author on reasonable request. Source data are provided with this paper.

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

## Acknowledgements
This work was supported by a grant from the Bavarian Ministry of Science and Arts.

## Author contributions
M.M. conceived and supervised the study. K.R. and M.M. analyzed the data. T.S., S.C., J.G., C.M., C.T., J.-N.S., F.W., S.Ba., and S.Bu. performed all experiments. M.V.M. provided patient data. M.L., D.L., B.G., F.Ha., F.Hi., and B.S. provided patient samples and clinical data. H.P. obtained ethical approval for the study, M.K. provided healthy control samples. E.O. and R.B. provided laboratory values and organized sample distribution. T.N., C.B., and A.G. organized biobanking of PBMCs. M.M. and K.R. drafted the manuscript.

## Funding

## Competing interests
The authors declare no competing interests.
