## [Peer Review File · Nature Communications]

REVIEWER COMMENTS

Reviewer #1 (Remarks to the Author):

In this manuscript, Renner et al present evidences that patients with severe SARS-COV2 infection present a T cell functional inactivation to anti-CD3 mediated activation.

They showed that anti-CD3 antibodies added to whole blood or PBMC induce less level of activation of monocytes and basophils (whole blood test) and less production of IFN-gamma and GM-CSF in supernatants of PBMC of severe Covid-19 patients.

The authors are also showing that this defect of T cell activation was reversible after disease resolution.

The observation is interesting and performed in a robust number of patients . However there are no attempts to try to decipher the mechanisms of T cell reduced level of response to anti-CD3 activation , the number of subjects studies longitudinally is low and the experiments designed to demonstrate that IL-2 reverts the state of “ T cell low response “ are extremely weak.

Major point:

a) The title of the paper is “T cell anergy in Covid-19 reflects virus-persistence...” which implies that the T cell functional inactivation is due to “anergy” and that such anergy is somehow mediated by the presence of the virus. T cell anergy is defined in immunology textbook as a form of functional inactivation of T cells mediated by antigen encounter and overcome by the addition of IL-2.

The authors are indeed showing that T cells are functionally inactivated during severe COVID-19 by it is very unlikely that such T cell functional inactivation was mediated by antigen encounter since the functional inactivation is present in all the T cells tested. Activation mediated by anti-CD3 (which is the only activation method tested in this manuscript) trigger all T cells, not just T cells specific for SARS-COV2. Since it is very unlikely that 100% of the T cells present in the patients are SARS-COV2 specific it is more likely that the functional inactivation observed is mediated by soluble factors (IL-10?) or by the lack of important AA necessary to T cell activation (I.e Arginase is released in many form of tissue damage, arginase depletes the AA Tryptofan , without Thriptofan T cells do not work). In other words, even though the data are showing that there is a level of functional inactivation of T cells, the claim that this is due to T cell anergy is not supported by any data. The restoration of IL-2 function showed in Suplemmentary figure 5 is practically not existent and the correlation between antigen persistence and T cell anergy is not demonstrated since I cannot see any data related to virus persistence / clearance in relation to T cell analysis.

I think the authors should use these interesting preliminary data to analyze the real cause of such T cell functional inactivation. T cells should be activated with different methods (i.e. PMA+ IONO, antigens like tetanous toxoid or other common antigens) and the ability of the sera of patients to inhibit T cell response should be also analyzed.

Reviewer #2 (Remarks to the Author):

In their article, Renner et al. study the correlation between the development of anergy in T cells and the outcome of COVID19 patients, classified by the need of ventilation or not and by their survival. Authors define anergy by studying the indirect effect of T cell activation on the expression of CD131 on basophils, CD123 on monocytes and CD11b on neutrophils. They established that these markers are modulated on the cell type of interest by cytokines that can be produced by T cells. The main finding is that in advanced form of the disease, i.e. for patients that require to be ventilated, these markers are less modulated, suggesting an increased anergy of T cells in these patients. The authors also noticed a difference in monocytes response between male and female. They also established a score using their marker and the proportion of basophils in the blood of patients to predict their outcome. The article is well written, the results are clear and well-presented but the conclusions drawn are not sufficiently backed up by the data shown. Furthermore, the overall impact of this potential increased anergy for patient with bad prognosis is not obvious to me, as these patients already show more elevated inflammation. I recommend to revise heavily this study. The authors claim that the anergy of T cells, and not of other cell types studied here (basophils, pDCs, Monocytes, Neutrophils) correlates with patient's outcome during COVID19. However, this point is not established. The authors showed that upon stimulation with anti-CD3 there is a decrease in GM-CSF, but not in IL-3 or IFN γ , in the supernatant from PBMC when comparing ventilated and non-ventilated patients. It is not sufficient to state that T cells from ventilated patients are anergic. Furthermore, the healthy control should be added in the figure. The ventilated "dead" patients are too few to draw any conclusion.

The second type of experiment performed by the authors to address this point is to block IL-3 production with anti-IL3 during anti-CD3 stimulation. It shows that the effect mediated by anti-CD3 is at least in part mediated by IL-3. However, it does not differentiate between anergy of T cells or lack of responsiveness of the other cell types to IL-3. And as IL-3 is produced to similar level in non-ventilated and ventilated patient after anti-CD3 stimulation (Suppl. Fig 5), this set of data is quite confusing. To formally address this point, the authors should:

- Look at Erk phosphorylation in T cells after stimulation in healthy, ventilated or non-ventilated patients. This can be done by flow cytometry and will really prove the anergic state of T cells.
- Stimulate whole blood cells with the various cytokines inducing a response in Suppl. F1 and compare the response in healthy, ventilated or non-ventilated patients. This will exclude or validate a lack of responsiveness of the "target cells".

If the T cells are really more anergic in ventilated COVID patients, the author should test whether the anergy is selective to CD4 or CD8 T cells or not.

Anergy is typically induced upon continuous TCR stimulation of T cells. If the T cells from COVID patients are more anergic, the concerned population should be specific for the SARS-Cov-2. Is there a difference in proportion of the virus specific population between healthy, ventilated, and non-ventilated patients? Are the alleged anergic T cells specific for viral epitopes or is it a general phenomenon?

In the discussion, the authors mention a low expression of CD11b on severely ill and ventilated patients. However, in figure S2 and S3, it appears that the basal level of expression of CD11b is higher in non-ventilated and in ventilated patients compared to healthy donors. Could the authors be more precise on this point? Can they explain the difference in basal expression?

They also mention the potential role of IL-3 and GM-CSF to explain the lower count in basophils and pDCs. This is an interesting point, but the authors should confirm this claim by

measuring these cytokines in the serum of the various type of patients.
The authors proposed that overcoming anergy would be a way to fight more efficiently the disease. They should propose treatment option for that in the discussion.

Reply to Reviewer #1 (Remarks to the Author):

In this manuscript, Renner et al present evidences that patients with severe SARS-COV2 infection present a T cell functional inactivation to anti-CD3 mediated activation.

They showed that anti-CD3 antibodies added to whole blood or PBMC induce less level of activation of monocytes and basophils (whole blood test) and less production of IFN-gamma and GM-CSF in supernatants of PBMC of severe Covid-19 patients.

The authors are also showing that this defect of T cell activation was reversible after disease resolution.

The observation is interesting and performed in a robust number of patients. However there are no attempts to try to decipher the mechanisms of T cell reduced level of response to anti-CD3 activation , the number of subjects studies longitudinally is low and the experiments designed to demonstrate that IL-2 reverts the state of “ T cell low response “ are extremely weak.

Major point:

a) The title of the paper is “T cell anergy in Covid-19 reflects virus-persistence...” which implies that the T cell functional inactivation is due to “anergy” and that such anergy is somehow mediated by the presence of the virus. T cell anergy is defined in immunology textbook as a form of functional inactivation of T cells mediated by antigen encounter and overcome by the addition of IL-2.

The authors are indeed showing that T cells are functionally inactivated during severe COVID-19 by it is very unlikely that such T cell functional inactivation was mediated by antigen encounter since the functional inactivation is present in all the T cells tested. Activation mediated by anti-CD3 (which is the only activation method tested in this manuscript) trigger all T cells, not just T cells specific for SARS-COV2. Since it is very unlikely that 100% of the T cells present in the patients are SARS-COV2 specific it is more likely that the functional inactivation observed is mediated by soluble factors (IL-10?) or by the lack of important AA necessary to T cell activation (I.e Arginase is released in many form of tissue damage, arginase depletes the AA Tryptophan , without Thryptofan T cells do not work). In other words, even though the data are showing that there is a level of functional inactivation of T cells, the claim that this is due to T cell anergy is not supported by any data. The restoration of IL-2 function showed in Suplementary figure 5 is practically not existent and the correlation between antigen persistence and T cell anergy is not demonstrated since I cannot see any data related to virus persistence / clearance in relation to T cell analysis.

Reply to “IL-2 and major point a”:

We thank the review for the valuable comments and agree that the term „anergy“ as defined in textbooks is indeed misleading, as the functional inactivation / hyporeactivity does not only affect the virus-specific T cells but the majority of T cells, as shown by polyclonal activation of T cells with anti-CD3. We thus have replaced “T cell anergy” with “T cell hyporeactivity” throughout the manuscript. This general T cell hyporeactivity is in our view the most interesting finding in our manuscript. It is already detectable in mild COVID-19 disease and most pronounced in severe forms of COVID-19.

As suggested by the reviewer we have analyzed additional patients to decipher the mechanisms of T cell hyporeactivity in a more robust manner:

a) Role of IL-2, IL-10 and L-tryptophan:

Our previous data with IL-2 were indeed weak. Analysis of IL-2 was not included in our original protocol with whole blood. When we decided to test IL-2 to overcome T cell hyporeactivity, the first wave of COVID-19 was over and fresh whole blood samples were no

longer available at our University Hospital. We therefore used frozen PBMCs from COVID-19 patients. However, as described in the manuscript, T cell hyporeactivity was difficult to detect with frozen PBMCs and the effects of IL-2 were thus low.

We now have resumed our analysis of fresh whole blood samples and used 10 samples from healthy controls, 13 from non-ventilated and 19 from ventilated patients. Whole blood samples were cultured with medium alone, with anti-CD3 or with anti-CD3 plus recombinant IL-2 (20 ng/ml) for 24h. Addition of recombinant IL-2 markedly improved T cell reactivity in non-ventilated patients, but was less effective in ventilated patients (new Supplementary Fig. 6). The previous IL2 – PBMC data are not shown any more.

We also analyzed the role of IL-10 for T cell hyporeactivity. First, we measured the IL-10 levels in unstimulated and anti-CD3-stimulated whole blood samples of COVID-19 patients and healthy controls. Although some COVID-19 patients showed a high release of IL-10 in the supernatant of T cell-activated whole blood, the difference between healthy controls and patients was not significant (see new Fig. 2 f). In order to functionally analyze the role of IL-10, we added a blocking anti-IL-10 antibody to whole blood stimulation assays of 13 ventilated COVID-19 patients (see new Suppl. Fig. 7 a, b). Blockade of IL-10 did not improve T cell stimulation in patients with COVID-19, indicating that IL-10 is not responsible for the reduced T cell reactivity in COVID-19 patients.

The role of IDO and tryptophan-depletion was analyzed by addition of excess amounts of L-tryptophan to override IDO effects. Addition of L-tryptophan had no beneficial effects on T cell activation, indicating that this pathway is not responsible for the reduced T cell reactivity in COVID-19 patients (see new Suppl. Fig. 7 c, d).

... “the correlation between antigen persistence and T cell anergy is not demonstrated since I cannot see any data related to virus persistence / clearance in relation to T cell analysis.”

Reply: The data related to virus persistence and T cell hyporeactivity are shown in Suppl. Tab. 2. We now have added a new figure with individual data points that shows that patients with a virus persistence above 15 days have a significantly longer duration of T cell hyporeactivity than patients with virus persistence below 15 days (Suppl. Tab. 2)

I think the authors should use these interesting preliminary data to analyze the real cause of such T cell functional inactivation. T cells should be activated with different methods (i.e. PMA+ IONO, antigens like tetanus toxoid or other common antigens) and the ability of the sera of patients to inhibit T cell response should be also analyzed.

Reply: As suggested by the reviewer we used PMA/Ionomycin to activate T cells in whole blood samples. In a series of pilot experiments we found that incubation of whole blood samples with PMA/Ionomycin for 6h or 24h is rather toxic to responder cells like basophils, monocytes and neutrophils. We thus could only use a 3h stimulation of whole blood samples with PMA/Ionomycin. This stimulation was performed in 11 healthy controls, 5 non-ventilated and 7 ventilated COVID-19 patients. Samples were stimulated either with PMA/Ionomycin or with PMA/Ionomycin + anti-IL3 in order to find out whether the PMA/Ionomycin-induced downregulation of CD131 on basophils is mediated by IL-3. With PMA/Ionomycin we found only a weak downregulation of CD131 on basophils which was not dependent on IL-3, suggesting that PMA/Ionomycin has a direct effect on basophils and that the downregulation of CD131 is not induced by release of IL-3 from activated T cells. Also the upregulation of CD123 on monocytes and CD11b on neutrophils was rather weak with PMA/Ionomycin,

preventing a meaningful analysis of downstream effects of T cell activation on responder cells. Please see figure below (not shown in the paper).

PMA/Ionomycin stimulation

100 μ l of heparinized whole blood from 11 healthy controls (Healthy), 5 non-ventilated (Non-vent.) and 7 ventilated (Vent.) COVID-19 patients was diluted with 200 μ l RPMI 1640 medium with and without PMA/Ionomycin (4 μ M / 67 μ M, Biolegend) and incubated at 37°C for 3 hours. Where indicated 10 μ g/ml of a blocking anti-IL-3 antibody was added. Expression of indicated surface markers was quantified by flow cytometry on basophils, CD14+ monocytes and neutrophils. Values depict the ratio of surface marker expression with PMA/Ionomycin or PMA/Ionomycin plus anti-IL-3 to surface marker expression with medium alone. Each sample is represented by one dot and the mean is marked in red (PMA/Ionomycin) or green (PMA/Ionomycin+anti-IL-3). One-way ANOVA with Bonferroni multiple comparison test was used. (* p<0.05, ** p<0.01, *** p<0.001)

... and the ability of the sera of patients to inhibit T cell response should be also analyzed.

Reply: We thank the reviewer for having addressed this important point. In order to find out whether plasma factors are responsible for T cell hyporeactivity in COVID-19 patients, we exposed washed whole blood from a healthy donor with plasma from 10 healthy donors, 13 non-ventilated and 19 ventilated COVID-19 patients. Plasma from non-ventilated patients and even more ventilated patients markedly suppressed T cell activation, as shown by a much lower downregulation of CD131 on basophils and a much lower upregulation of CD123 on monocytes and CD11b on neutrophils (see new Fig. 3). The suppressive effects of COVID-19 plasma could not be explained by immunosuppressive medication, as only 23% of the non-ventilated and 33% of the ventilated patients received steroids. No other immunosuppressive agents were given to any of the patients. In Fig. 3 the plasma samples of patients with steroids are marked in blue and show no increased T cell suppression.

Reply to Reviewer #2 (Remarks to the Author):

In their article, Renner et al. study the correlation between the development of anergy in T cells and the outcome of COVID19 patients, classified by the need of ventilation or not and by their survival. Authors define anergy by studying the indirect effect of T cell activation on the expression of CD131 on basophils, CD123 on monocytes and CD11b on neutrophils. They established that these markers are modulated on the cell type of interest by cytokines that can be produced by T cells. The main finding is that in advanced form of the disease, i.e for patients that require to be ventilated, these markers are less modulated, suggesting an increased anergy of T cells in these patients. The authors also noticed a difference in monocytes response between male and female. They also established a score using their marker and the proportion of basophils in the blood of patients to predict their outcome.

The article is well written, the results are clear and well-presented but the conclusions drawn are not sufficiently backed up by the data shown. Furthermore, the overall impact of this potential increased anergy for patient with bad prognosis is not obvious to me, as these patients already show more elevated inflammation. I recommend to revise heavily this study.

Reply: We thank the reviewer for the valuable comments to improve our manuscript. We have addressed in the discussion the apparent discrepancy that the most severely affected COVID-19 patients have the most severe T cell hyporeactivity but a pronounced systemic inflammation. We show that severely affected COVID-19 patients were unable to clear the virus for prolonged periods of time and that the duration of T cell hyporeactivity correlated with virus persistence (Suppl. Tab. 2). Thus elevated inflammation in severely ill COVID-19 patients can not be triggered by T cells but is most likely triggered by the virus and / or persistent hyper-activation of other cells types (e.g. innate cells like monocytes).

The authors claim that the anergy of T cells, and not of other cell types studied here (basophils, pDCs, Monocytes, Neutrophils) correlates with patient's outcome during COVID19. However, this point is not established. The authors showed that upon stimulation with anti-CD3 there is a decrease in GM-CSF, but not in IL-3 or IFN γ , in the supernatant from PBMC when comparing ventilated and non-ventilated patients. It is not sufficient to state that T cells from ventilated patients are anergic. Furthermore, the healthy control should be added in the figure. The ventilated "dead" patients are too few to draw any conclusion.

Reply: This point of the reviewer relates to the assays with PBMCs shown now in Suppl. Fig. 8. We would like to point out that T cell hyporeactivity is only detectable in whole blood samples of COVID-19 patients but not or to a much lower degree in assays with frozen PBMCs. In the assays with whole blood there is a large difference between non-ventilated and ventilated COVID-19 patients, as T cell activation in ventilated COVID-19 patients results in decreased downstream effects on basophils, monocytes and neutrophils (Fig. 1), and a markedly decreased release of IL-3, GM-CSF and IFN-gamma from T cells (Fig. 2 c-e). In contrast, with frozen PBMCs almost no differences between non-ventilated and ventilated COVID-19 patients could be detected, as we only observed a somewhat lower release of GM-CSF in ventilated patients, but no differences for upregulation of CD25, for release of IL-3 and IFN-gamma or for downstream activation of basophils and monocytes (now Suppl. Fig. 8). Therefore we have abandoned the analysis of PBMCs and have not included healthy volunteers in the assays with PBMCs. As the number of PBMC samples from dead ventilated patients are low we have removed this category from Suppl. Fig. 8 and now just show non-ventilated and ventilated patients (including "survived" and "dead").

The second type of experiment performed by the authors to address this point is to block IL-3 production with anti-IL3 during anti-CD3 stimulation. It shows that the effect mediated by anti-CD3 is at least in part mediated by IL-3. However, it does not differentiate between anergy of T cells or lack of responsiveness of the other cell types to IL-3. And as IL-3 is produced to similar level in non-ventilated and ventilated patient after anti-CD3 stimulation (Suppl. Fig 5), this set of data is quite confusing.

Reply: The reviewer seems to confuse the results of the assays with whole blood and with PBMCs. Only in assays with PBMCs, IL-3 is produced to similar levels in non-ventilated and ventilated patients after anti-CD3 stimulation and there was also no difference in the response of basophils to anti-CD3 in non-ventilated and ventilated patients (now Suppl. Fig. 8). In contrast, in assays with whole blood IL-3 is produced at about 5-fold lower levels in ventilated compared to non-ventilated patients after anti-CD3 stimulation (Fig. 2 c) and consistently there is also a pronounced difference in the response of basophils to anti-CD3 (Fig. 1 a).

In order to show directly that downstream responder cells like basophils, monocytes and neutrophils are not hypo-responsive in COVID-19 patients, we incubated whole blood samples of 10 healthy controls, 13 non-ventilated, and 15 ventilated patients with anti-CD3 or just with recombinant IL-3 and GM-CSF for 24h (new Supplementary Fig. 5). While the response of basophils, monocytes and neutrophils to T cell activation with anti-CD3 was very low in COVID-19 patients, their response to recombinant IL-3 and GM-CSF was fully preserved in a cell type specific manner (preferential response of basophils to IL-3, of monocytes to IL-3 and GM-CSF and of neutrophils to GM-CSF). This argues against an anergy of responder cells.

To formally address this point, the authors should:

- Look at Erk phosphorylation in T cells after stimulation in healthy, ventilated or non-ventilated patients. This can be done by flow cytometry and will really prove the anergic state of T cells

Reply: We acknowledge that Erk phosphorylation would be suitable to measure T cell activation. We have chosen to measure the release of cytokines and the downstream effects of cytokines, as cytokine release is one of the most important functional activities of T cells. Erk phosphorylation is an intermediate step in a complex signal transduction pathway and only an indirect readout of T cell function. In addition, flow cytometric analysis of Erk phosphorylation would require intracellular staining that is difficult in whole blood samples. We therefore have chosen to not analyze Erk phosphorylation.

- Stimulate whole blood cells with the various cytokines inducing a response in Suppl. F1 and compare the response in healthy, ventilated or non-ventilated patients. This will exclude or validate a lack of responsiveness of the “target cells”.

Reply: In order to show directly that downstream responder cells like basophils, monocytes and neutrophils are not hypo-responsive in COVID-19 patients, we incubated whole blood samples of 10 healthy controls, 13 non-ventilated, and 15 ventilated patients with anti-CD3 or just with recombinant IL-3 and GM-CSF for 24h (new Supplementary Fig. 5). While the response of basophils, monocytes and neutrophils to T cell activation with anti-CD3 was very low in COVID-19 patients, their response to recombinant IL-3 and GM-CSF was fully preserved in a cell type specific manner (preferential response of basophils to IL-3, of monocytes to IL-3 and GM-CSF and of neutrophils to GM-CSF). This argues against an lack of responsiveness of the “target cells”.

If the T cells are really more anergic in ventilated COVID patients, the author should test whether the anergy is selective to CD4 or CD8 T cells or not.

Reply: Fig. 2 c, d, e shows that stimulation of T cells with anti-CD3 does not induce any release of IL-3, GM-CSF or IFN-gamma in ventilated COVID-19 patients, while there is a marked release in non-ventilated COVID-19 patients and healthy controls. As IL-3, GM-CSF and IFN-gamma are released from both CD4+ and CD8+ T cells we can assume that T cell hyporeactivity affects both CD4+ and CD8+ T cells. We have included this in the results section. Moreover, separation of CD4+ and CD8+ T cells would require a lot of manipulation of the blood samples or intracellular staining that is difficult in whole blood assays.

Anergy is typically induced upon continuous TCR stimulation of T cells. If the T cells from COVID patients are more anergic, the concerned population should be specific for the SARS-Cov-2. Is there a difference in proportion of the virus specific population between healthy, ventilated, and non-ventilated patients? Are the alleged anergic T cells specific for viral epitopes or is it a general phenomenon?

Reply: We agree that the term “anergy” is misleading, as it suggests that T cells are hyporesponsive because they have been continuously stimulated with viral antigens. We have used polyclonal activation of T cells with immobilized anti-CD3 that activates all T cells and not only virus-specific T cells. Therefore the T cell hyporeactivity is not specific for viral epitopes but a more general phenomenon. We thus have replaced the term “T cell anergy” with “T cell hyporeactivity” throughout the manuscript.

In addition, we found out the reduced T cell activation in whole blood samples of patients with COVID-19 is T cell extrinsic and caused by factors present in the plasma of COVID-19 patients. We exposed washed whole blood from a healthy donor with plasma from 10 healthy donors, 13 non-ventilated and 19 ventilated COVID-19 patients. Plasma from non-ventilated patients and even more ventilated patients markedly suppressed T cell activation, as shown by a much lower downregulation of CD131 on basophils and a much lower upregulation of CD123 on monocytes and CD11b on neutrophils (see new Fig. 3). The suppressive effects of COVID-19 plasma could not be explained by immunosuppressive medication, as only 23% of the non-ventilated and 33% of the ventilated patients received steroids. No other immunosuppressive agents were given to any of the patients. In Fig. 3 the plasma samples of patients with steroids are marked in blue and show no increased T cell suppression. Neither IL-10 nor tryptophan-degradation could explain the plasma-induced suppression of T cells in COVID-19 patients, as blockade of IL-10 with a blocking antibody or addition of excess L-tryptophan had no beneficial effects on T cell reactivity (new Supplementary Fig. 7).

In the discussion, the authors mention a low expression of CD11b on severely ill and ventilated patients. However, in figure S2 and S3, it appears that the basal level of expression of CD11b is higher in non-ventilated and in ventilated patients compared to healthy donors. Could the authors be more precise on this point? Can they explain the difference in basal expression?

Reply: Thank you for pointing this out. We have rewritten this part of the discussion as follows: “Compared to healthy controls, basal CD11b expression on neutrophils is higher in non-ventilated and somewhat higher in ventilated patients. This indicates that COVID-19 leads to an activation of neutrophils. The pronounced T cell hyporeactivity in ventilated

patients may explain why basal CD11b expression is significantly lower in ventilated compared to non-ventilated patients and again lower in patients with a fatal outcome.”

They also mention the potential role of IL-3 and GM-CSF to explain the lower count in basophils and pDCs. This is an interesting point, but the authors should confirm this claim by measuring these cytokines in the serum of the various type of patients.

Reply: Some years ago we have spent great efforts to measure IL-3 in the plasma of healthy volunteers. We used different IL-3 ELISAs and also a very sensitive functional bioassay (activation of basophils) to detect IL-3 in the plasma. While there were sometimes inconsistent signals with some ELISAs, we were never able to detect functionally active IL-3 in the plasma of healthy volunteers. We therefore have chosen not to measure IL-3 or GM-CSF in the plasma. Apart from this, we believe that basophil and pDC counts are not determined by plasma levels of IL-3 or GM-CSF but by the level of both cytokines in tissues (e.g. bone marrow, spleen).

The authors proposed that overcoming anergy would be a way to fight more efficiently the disease. They should propose treatment option for that in the discussion.

Reply: Our data show that IL-2 could help to overcome T cell hyporeactivity in COVID-19 patients (see new Supplementary Fig. 6 with whole blood). Apart from this, we currently work on the identification of the plasma factors that induce T cell hyporeactivity in COVID-19 patients. Once these factors are identified, their inhibition could be a valuable treatment option.

REVIEWER COMMENTS

Reviewer #1 (Remarks to the Author):

The authors improved the manuscript with the addition of new data derived from the study of more patients. The observation that soluble components in the plasma of severe COVID-19 patients can mediate a suppression of T cells is novel and might open a new area of investigation.

The paper introduction and discussion should be updated . New data published in the last couple of months have for example show that SARS-CoV2 specific T cells function is suppress only in patients with more severe diseases (see 1 Rydzynski Moderbacher C, et al. Antigen-Specific Adaptive Immunity to SARS-CoV-2 in Acute COVID-19 and Associations with Age and Disease Severity. Cell 2020; 183: 996–1012.e19. Tan AT, et al. Early induction of functional SARS-CoV-2-specific T cells associates with rapid viral clearance and mild disease in COVID-19 patients. Cell Reports 2021; 53: 108728–13.) these findings needs to be added and discussed since the are in line with the findings of this work.

In addition, he authors wrote:

“ Current data on T cell reactivity in COVID-19 are conflicting and indicate that patients with severe COVID-19 can have either insufficient^{20,21} or excessive²²⁻²⁴ T cell responses

This is a real oversimplification of the data currently available and the authors should explain better the differences between data reporting the profile of total T cells or data reporting the features of antigen-specific T cells. They should point out in their introduction and discussion that their analysis focus on total T cell function and not on SARS-CoV2 T cells and they should discuss the data available in relation to the severity of infection. A large quantity of data is available showing that patients with mild and aymptomatic infection have a perfectly normal T cell function and such data can be discussed.

Reviewer #2 (Remarks to the Author):

The authors answered most of my concerns. The change from “anergy” to “hyporeactivity” is removing a lot of confusions. Furthermore, they have addressed the potential hyporesponsiveness of target cells. I apologize for the confusion I made between PBMCs and whole blood data. The addition of the data concerning the direct effect of plasma on T cells renders the manuscript even more interesting. Obviously, I am curious to know which compounds is responsible of this effect. However, I think the paper can be published as it is.

Reviewer #1 (Remarks to the Author):

The authors improved the manuscript with the addition of new data derived from the study of more patients. The observation that soluble components in the plasma of severe COVID-19 patients can mediate a suppression of T cells is novel and might open a new area of investigation. The paper introduction and discussion should be updated. New data published in the last couple of months have for example show that SARS-CoV2 specific T cells function is suppress only in patients with more severe diseases (see 1 Rydzynski Moderbacher C, et al. Antigen-Specific Adaptive Immunity to SARS-CoV-2 in Acute COVID-19 and Associations with Age and Disease Severity. Cell 2020; 183: 996–1012.e19. Tan AT, et al. Early induction of functional SARS-CoV-2-specific T cells associates with rapid viral clearance and mild disease in COVID-19 patients. Cell Reports 2021; 53: 108728–13.) these findings needs to be added and discussed since the are in line with the findings of this work.

In addition, he authors wrote:

“ Current data on T cell reactivity in COVID-19 are conflicting and indicate that patients with severe COVID-19 can have either insufficient^{20,21} or excessive²²⁻²⁴ T cell responses

This is a real oversimplification of the data currently available and the authors should explain better the differences between data reporting the profile of total T cells or data reporting the features of antigen-specific T cells. They should point out in their introduction and discussion that their analysis focus on total T cell function and not on SARS-CoV2 T cells and they should discuss the data available in relation to the severity of infection. A large quantity of data is available showing that patients with mild and asymptomatic infection have a perfectly normal T cell function and such data can be discussed.

Reply to Reviewer #1:

We thank the reviewer for pointing out the two very interesting papers to us. We have now included and discussed both papers in the introduction and discussion as follows:

“Recently, impaired SARS-Cov-2 specific CD4+ and CD8+ T cell responses were found to be associated with older age and more severe outcome, while humoral immune responses were not predictive {Rydzynski Moderbacher, 2020 #67; Tan, 2021 #68}. Our data are in line with these findings and additionally suggest that critically ill COVID-19 have a more general T cell suppression that could also impair development of SARS-CoV-2 specific T cell responses. Many critically ill COVID-19 patients suffer from fungal infections, which may be caused by the general T cell suppression in these patients {Evert, 2021 #72; Kuehn, 2020 #70}.

We also have made clear at several text passages that we have analyzed polyclonal and not SARS-CoV-2 specific T cell activation. Our analysis only included hospitalized (non-ventilated and ventilated) patients but no non-hospitalized patients with mild or asymptomatic infection. As hospitalized non-ventilated patients have only a weak impairment of T cells function (compared to ventilated patients), we consider it likely that asymptomatic patients could have a completely normal T cell function.

Reviewer #2 (Remarks to the Author):

The authors answered most of my concerns. The change from “anergy” to “hyporeactivity” is removing a lot of confusions. Furthermore, they have addressed the potential hyporesponsiveness of target cells. I apologize for the confusion I made between PBMCs and whole blood data. The addition of the data concerning the direct effect of plasma on T cells renders the manuscript even more interesting. Obviously, I am curious to know which compounds is responsible of this effect. However, I think the paper can be published as it is.

Reply: We thank the reviewer for the positive feedback.